# A Qualitative Content Analysis of Rural and Urban School Students’ Menstruation-Related Questions in Bangladesh

**DOI:** 10.3390/ijerph191610140

**Published:** 2022-08-16

**Authors:** Deena Mehjabeen, Erin C. Hunter, Mehjabin Tishan Mahfuz, Moshammot Mobashara, Mahbubur Rahman, Farhana Sultana

**Affiliations:** 1Translational Health Research Institute, Western Sydney University, Penrith, NSW 2751, Australia; 2School of Public Health, Faculty of Medicine and Health, The University of Sydney, Sydney, NSW 2006, Australia; 3Department of International Health, Johns Hopkins Bloomberg School of Public Health, Baltimore, MD 21205, USA; 4Environmental Interventions Unit, Infectious Diseases Division, International Centre for Diarrhoeal Disease Research, Bangladesh (ICDDR,B), Dhaka 1212, Bangladesh or; 5Public Health & Preventive Medicine, Monash University, Melbourne, VIC 3004, Australia

**Keywords:** qualitative research, menstrual health, menstrual hygiene management, Bangladesh, schools, puberty, health education

## Abstract

Nearly half of Bangladeshi girls reach menarche without knowledge of menstruation and many fear seeking support due to pervasive menstrual stigma. We aimed to explore the types of menstruation-related information and support adolescent female and male students want but may be uncomfortable verbalising. We installed a locked box in four school classrooms in rural and urban Bangladesh as part of a menstrual hygiene management pilot intervention between August 2017 and April 2018. Trained teachers provided puberty education to female and male students in classes 5–10 (ages 10–17 years) and encouraged students to submit questions anonymously to the boxes if they did not want to ask aloud. We conducted a content analysis of the 374 menstruation-related questions from a total of 834 submissions. Questions regarded experiences of menstrual bleeding (35%); menstrual symptoms and management (32%); menstrual physiology (19%); behavioural prescriptions and proscriptions (6%); concerns over vaginal discharge (4%); and menstrual stigma, fear, and social support (4%). Students wanted to understand the underlying causes of various menstrual experiences, and concern over whether particular experiences are indicative of health problems was pervasive. Ensuring comprehensive school-based menstruation education and strengthening engagement among schools, parents, and healthcare providers is important for improving access to reliable menstrual health information and may relieve adolescents’ concerns over whether their menstrual experiences are ’normal’.

## 1. Introduction

Adolescence is a pivotal time of physical, mental, emotional, and social change. For some, the onset of puberty and menstruation in adolescent girls signals marriage readiness and brings socio-cultural expectations to leave school, shoulder more household responsibilities, and explore new sexual experiences [1]. The onset of puberty is also a phase where girls may experience negative feelings about their bodies and low self-esteem [2]. Along with family expectations, girls reaching puberty must also navigate the onset of menstruation and manage menstrual experiences at home and in public, such as in schools. With a global increase in the number of girls enrolling in primary education and transitioning into high school, often in co-educational settings, there is a need for support regarding menstrual care among adolescent girls [3,4].

Timely and accurate knowledge about menstruation is important for girls to have positive menstrual experiences, along with supportive social and physical environments and access to sanitation facilities and menstrual materials [5,6]. However, menstrual stigma and proscriptions on open discussion of the topic restrict girls’ access to knowledge about menstruation. Across low- and middle-income countries (LMICs), girls have limited knowledge and understanding about menstruation before reaching menarche [7]. In Bangladesh, a study among 12 to 16-year-olds found that at least 64% of girls surveyed reported feeling scared upon reaching menarche [8,9]. Mothers interviewed in the same study described the topic of menstruation as irrelevant for discussion with premenarcheal girls and that it might make their daughters ‘unnecessarily precocious’ [8] (p. 199). A cross-sectional study conducted in Bangladesh among 106 female nursing students aged 20 years and below found that most (67%) were surprised during their first menstrual cycle [10]. The neighbouring nation of India—which shares some cultural and social norms with Bangladesh—reported a cross-sectional study in two girls’ schools in the state of West Bengal, in which approximately 38% of girls in an urban school were aware of menstruation before menarche while only 21% in a rural school were; and most girls had no idea about the cause of menstruation [11]. By contrast, in a study for school-going girls in urban Turkey, all participants (100%) were aware of menstruation prior to menarche [12]. In the United Kingdom, a survey by Plan International found that only 14% of girls aged 14–21 years did not know about periods before they reached menarche [13].

Poor menstrual hygiene often leads to negative psychosocial and physical health outcomes [14]. Studies conducted in Africa (Ethiopia, Kenya, Ghana, The Gambia, and Tanzania) and Asia (Cambodia) reported negative psychosocial outcomes involving feelings of distress, fear, shame, and anxiety, which have been linked with difficulties in managing menstrual bleeding and school absenteeism [15,16,17]. Although there is limited evidence, systematic reviews for LMICs [14,18,19,20], and quantitative studies conducted in India [21,22], Ethiopia [23], and Rwanda [24] reported the negative health outcomes associated with poor menstrual hygiene management such as vaginal discharge and odour, urogenital infections, reproductive tract infections (RTIs), anaemia, and even infertility. Addressing these health outcomes typically involves the implementation of ‘hardware interventions’ requiring physical or material resources such as menstrual materials (e.g., menstrual cups, reusable or disposable sanitary pads, etc.) and infrastructure—such as water, sanitation, and hygiene (WASH) or waste disposal facilities [14,18,25]. Psychosocial wellbeing outcomes are addressed through ‘software interventions’ such as programs that focus on education to improve menstrual knowledge, especially for girls already in education [19,25,26], and other programs that address harmful stigma to improve menstrual practices and prepare girls for menstruation. Comprehensive menstruation education can support effective menstrual practice, identify abnormalities, and equip girls to become more confident and informed—and to have agency to care for their bodies [14].

We co-designed an intervention package with stakeholders (including the Ministry of Education, NGOs, and schoolteachers and authorities) intended to create more supportive school environments for menstruating students and improve menstrual knowledge, practices, comfort, and school attendance. The multi-component intervention was designed based on participatory formative research with school students and included parent and school management committee engagement, schoolteacher training to provide more comprehensive and engaging puberty education, distribution of puberty information booklets and menstrual hygiene packs to students, improvements to school sanitation facilities, and creation of school gender committees and a national-level advisory group. The intervention was piloted and evaluated from August 2017 to April 2018 in four urban and rural schools in Bangladesh. The effects of the intervention on primary outcomes are presented separately [27]. This paper is a sub-study that aimed to explore the breadth of information regarding menstruation that Bangladeshi adolescents sought from their schoolteachers during the intervention pilot, demonstrating the types of knowledge they desired yet had been unable to reliably access.

## 2. Methods

### 2.1. Study Context

The study took place in two schools in the capital city of Dhaka and two schools in rural areas of Manikganj District, Bangladesh. The methods of selecting study schools have been described previously [28]. Three of the four schools were non-government schools, and one was entirely a government (public) school. All were co-educational schools; however, boys and girls attended the two urban schools at different time shifts and in rural schools, boys and girls attended classes separately. The total number of students (classes 3–10) at the urban and rural schools was 5080, with 3140 in the two urban schools and 1940 in the two rural schools. The study population included all girls and boys in classes 5–10 (ages 10–17 years), studying at the selected schools.

### 2.2. Description of the ‘Question Box’ Intervention Component

Study team members comprised of male and female researchers with expertise in health education, medicine, anthropology, and psychology provided schoolteachers 5-days training (3 days of initial training and a 2-day ‘refresher’ training mid-intervention) on how to deliver a participatory puberty and menstruation education curriculum. It was part of a broad multi-component pilot intervention to promote a supportive school environment for menstruating students in Bangladesh [28]. The curriculum covered physical, mental, and emotional changes during puberty—with particular emphasis on menstrual health and hygiene—since current textbooks exclude information on the practical aspects of menstrual self-care. Teachers delivered the curriculum to their students, separately by sex, in multiple sessions over a six-month period [28]. The research team installed a wooden question box (Figure 1) with a slit in the top and a lock-and-key in each classroom where teachers conducted puberty education sessions. Students were encouraged to anonymously submit any puberty-related questions if they felt uncomfortable asking aloud during class. Students had direct access to the Question Box and could submit questions when teachers or others were not present in the classroom. Teachers periodically reviewed these questions and had been trained to provide answers based on information in their instructors’ manual or from consultation with physicians on the research team in cases where they did not feel confident in their own knowledge. In two schools, teachers asked the study team to provide answers directly to female students in a half-hour session toward the end of the intervention. The study team also provided a list of written answers to the most common questions to all teachers and students after conclusion of the study. The participants knew the names and occupations of the researchers and reason for studying the pilot intervention; and there was no relationship between researchers and the student participants prior to beginning the study.

### 2.3. Data Collection and Analysis

The study team collected all questions submitted to the question boxes over the 6-month pilot and entered them in Bengali (local language) into a spreadsheet according to school and class level. We took an inductive approach to analyse the content of the submitted questions, focusing on menstrual health and hygiene. Since the sub-study aim was to explore the type of information students wanted to know about menstruation, qualitative content analysis was the most appropriate method of data analysis. We also took a number of steps to ensure trustworthiness according to the four criteria (credibility, dependability, confirmability, and transferability) by Lincoln and Guba [29]. D.M. and E.C.H. familiarised themselves with the data by reading through the submissions multiple times. The study team held analytic meetings to discuss preliminary ideas and main points that the participants were expressing [30]. Each submitted question, labelled with the school and class level it came from, was considered a textual unit and D.M. conducted initial open coding of the Bengali questions in the spreadsheet. The first two authors then printed the spreadsheet and cut and hand-sorted the textual units to obtain a complete visual presentation of the data. This process aided the next steps of our iterative approach to indexing the dataset into meaningful categories and sub-categories and identifying cross-cutting themes.

We began by sorting the textual units into initial groups based largely on commonality in their manifest content (aided by, but not restricted to, our initial codes) [31]. As we moved through the analysis, we identified that multiple groupings of questions were related to each other at a higher order, and thus formed a number of sub-categories which were sorted into a broader category. We also noted that some of the large categories could be further divided into smaller sub-categories. Thus, we created the categories and sub-categories after performing different levels of abstraction. During the process of categorising all textual units into mutually exclusive categories, we began to identify two main cross-cutting themes that were evident within and across multiple categories. These themes expressed the more latent content of many of the submitted questions [31], in other words highlighting the underlying ‘essence’ of the data [32] (p. 727). We co-wrote memos documenting our analytic decisions during the hand sorting—recording any changes in the sorting decisions and tracking our recoding and relabelling when necessary. The other co-authors provided additional interpretation of the findings, thus maintaining the trustworthiness (dependability and confirmability) of the analysis process [31,33,34,35]. D.M. then translated all questions into English for presentation in the paper.

## 3. Results

Out of 834 submitted puberty-related questions, around 45% (374) were related to menstruation, the focus of this analysis. The remaining questions on other aspects of puberty are beyond the scope of analysis for this paper. Thirty-five percent of the 374 questions were about experiences of menstrual bleeding, 32% about menstrual symptoms, 19% about other menstrual physiology, 6% about behavioural prescriptions and proscriptions during menstruation, 4% about concerns regarding vaginal discharge, and 4% related to menstrual stigma, fear, and social support (Table 1). Out of the 374 menstruation-related questions, 175 were from urban schools, and 199 were from rural schools. The numbers of submissions in the urban and rural schools are presented separately by class levels (Table 2). Exemplars of questions are presented in Table 3 and further details can be found in the Appendix A.

### 3.1. Experiences of Menstrual Bleeding

Nearly a third of the questions were about experiences of menstrual bleeding, including the timing of menstrual periods and heaviness of blood flow. Classes 7 and 8 in rural schools and class 9 in urban schools had more of these questions as compared to other class levels. Questions indicated concerns over whether specific experiences are ‘normal’ or indicated health problems. Students also requested advice on managing menstrual bleeding hygienically. In the following sub-sections, we present findings across three sub-categories of questions related to menstrual bleeding.

#### 3.1.1. Timing of Menstruation

Questions related to the timing of menstruation were common, including both timing of menarche and regularity or frequency of menstrual bleeding thereafter. Most menarche-related questions were from urban schools while the rural school students asked many questions on menstrual irregularity. Students wanted information about the age of menarche, whether a missing or delayed period is a problem, whether it was normal to experience delays in starting menstruation after reaching puberty, and possible health problems if girls do not menstruate. Example questions include: ‘*If girls do not get their period, is it a health problem?*’, ‘*My age is 13 years but why hasn’t my period started?*’ and ‘*Will my irregular period become regular when I reach 15 years of age?*’.

Students sought clarity around reasons for perceived irregularities in their cycles after reaching menarche and whether it is a problem. Questions such as, ‘*Why is my period lasting for more than seven days?*’, ‘*Period occurs twice in a year but lasts for 30 days, is it okay?*’, ‘*Is it a health problem if girls have periods every 3–4 months, and will it be a problem for them to conceive?*’, and ‘*What is the remedy when we miss our monthly period?*’ were very common.

#### 3.1.2. Menstrual Blood Flow

Four percent of the menstruation-related questions, primarily from rural schools, dealt explicitly with the amount or changes in menstrual blood flow. Students wanted to know the reason for variation in the amount of blood they lost on any given day during a menstrual period, why some girls have ‘heavy flow’ and others have ‘light flow’ during their periods, and why some girls pass blood clots. In addition to wanting explanations for such occurrences, students also asked whether excessive bleeding could have consequences and asked for advice on what they should do to address excessive bleeding.

#### 3.1.3. Requests for Menstrual Hygiene Management Advice

Students demonstrated that they wanted to learn more about maintaining menstrual hygiene (i.e., managing their bleeding) and asked for specific recommendations. They also complained about inadequate sanitation facilities at school. A large proportion of requests for menstrual hygiene management advice was from urban high schools. Students wanted recommendations on which types of menstrual materials they should use and how often they should change to avoid negative consequences. For example, questions included, ‘*Is it good to use tissues* [to absorb menses] *during periods?*’, ‘*Should we always carry emergency sanitary pads in our school bag?*’, ‘*How often should we change pads or cloths?*’, and ‘*Is there a risk of infection if we use cloth?*’. They mentioned that school toilets remained shut and expressed a necessity for clean toilets with adequate water supply.

### 3.2. Menstrual Symptoms and Management

Roughly a third of the menstruation-related questions requested explanations for specific menstrual symptoms. Students from class 7 in rural schools and class 8 in both urban and rural schools submitted more of these questions as compared to other class levels. Students wanted to know why lower abdominal cramps, headaches, leg pain, acne, backaches, pelvic pain, body aches, swollen breasts, lack of appetite, and weakness occurred during menstruation. Beyond explanations for the causes of menstrual symptoms, students also expressed a need for more information on effective pain relief options for severe cramping and remedies for other menstrual symptoms, whether it was advisable to take medication for the pain, and solutions if they miss a menstrual period. Common questions included: ‘*Why do girls have abdomen cramps during menstruation?*’ and ‘*How can we decrease cramps during periods?*’. Questions such as ‘*Why can’t we take medication to reduce abdominal cramps during menstruation?*’ indicate some have been told not to use analgesics for menstrual pain.

### 3.3. Menstrual Physiology

Around 19% of the menstruation-focused questions were about other aspects of menstrual physiology. These were predominantly from classes 8 and 9 in urban schools and class 7 in rural schools. Students wanted to know why menstruation occurs and what it entails. For instance, ‘*What is the reason for menstruation?*’, and ‘*What happens during menstruation?*’ were commonly asked. From a biological perspective, menstruation, fertilisation, and reproduction are interrelated. There were queries regarding female reproductive organs, for example, ‘*What is ovum?*’ and the link between menstruation and reproduction, such as: ‘*Why can’t girls get pregnant without menstruation?*’. Students asked whether menstruation occurs in boys, people with disabilities, and transgender people. They were also curious to know whether only humans experience menstruation.

### 3.4. Behavioural Prescriptions and Proscriptions during Menstruation

In six percent of submitted questions, mostly from urban schools, students sought advice regarding appropriate behaviours during menstruation. Questions demonstrated students’ awareness of various cultural and religious proscriptions on the behaviour of menstruating girls and women, such as those regarding diet, limited movement and social interaction, and abstinence from sexual intercourse. Students wanted to know whether they should follow such behavioural recommendations and sought clarification about their purpose or benefit. For example, students asked ‘*Why abstain from sex during menstruation?*’, ‘*Why should we be cautious during menstruation?*’, ‘*Why aren’t you allowed to eat sour foods during menstruation?*’, and *“Why don’t our father and mother let us go outside during menstruation?*’. Other questions did not reference specific restrictions but rather asked for general advice on recommended foods and how girls should ‘conduct themselves’ during menstruation. For instance, ‘*What are the essential food items that girls should eat during menstruation?*’, and ‘*What should we do during our period?*’ were asked multiple times.

### 3.5. Concerns Regarding Vaginal Discharge

Four percent of the questions regarded experiences of vaginal discharge, which is a typical part of the normal menstrual cycle and may first be noticed 6 to 12 months before menarche. A large portion of the questions was from urban schools. Students asked what the discharge was, where it comes from, why it occurs, and were often keen to know whether their experiences of vaginal discharge were ‘normal’ or signs of a problem. Students were curious to know if excessive discharge could indicate the presence of a disease that requires treatment and wondered about the reasons for the discharge’s odour. Students also asked about the duration and the age at which vaginal discharge begins.

### 3.6. Menstrual Stigma, Fear, and Social Support

Students in urban schools submitted questions that reflected their fear of menarche and menstruation in general, as well as the presence of menstrual stigma and the need for more positive social support for menstruating girls. This category represented four percent of the menstruation-related submissions. The questions demonstrated students’ eagerness to address teasing from male and female peers and the emotional distress they experience related to menstruation. Questions such as, ‘*I’m fearful about menstruation, is there a way to mitigate this?*’ and ‘*Every girl gets periods, but why do girls make fun of menstruation?*’ exemplify this category. Questions also indicated a sense of remonstration about the lack of support from school authorities and other adults. Students communicated their need for a specially trained teacher at school to provide puberty and menstruation education. They also lodged complaints about having a poor classroom environment. Some also suggested that it is essential to discuss the information they received with their mothers, saying ‘*We talk about these things with our elder sister. I think we need to talk to our mothers about periods’.* One submission combined these sentiments in a plea for help:


*I think this is a problem for many girls. If we tell the adults, they do not want to listen. They don’t give us any appropriate answer. Many of us suffer from mental health issues due to this and it harms us. Please provide us with a way to come out of this mental suffering.*
—Student in Class VII, urban school

### 3.7. Cross-Cutting Themes

We identified two key themes that cut across the categories of questions. First, students wanted to know more than just *what* they can expect to occur in their bodies during menstruation, rather they wanted to understand the underlying causes or reasons for those experiences as well. This can be seen in the predominant framing of the questions as ‘*why* does [x] occur?’ and requests for more comprehensive explanations of the menstrual cycle and the underlying reasons for what they experience during menstruation. Secondly, concern or worry over whether particular menstrual experiences are ‘normal’ or indicative of health problems was pervasive.

## 4. Discussion

In our study, it was evident from the 834 questions participants submitted, that there was an appetite from students for the Question Box—particularly for asking about menstruation, which made up a large proportion of the submissions. The majority of the students’ questions concerned the experiences of menstrual bleeding and menstrual symptoms and their management. The remaining submissions involved questions on menstrual physiology, behavioural prescriptions and proscriptions, vaginal discharge, and menstrual stigma, fear and social support. There were more questions on menstrual symptoms and timing of menstruation from classes 7 and 8 in rural schools, which may indicate that these types of questions were especially of interest to students at the age when menarche has recently occurred or is expected.

The cross-cutting theme of students overwhelmingly framing questions such as ‘*why* does [x] occur?’ demonstrates their need for more comprehensive explanations of the menstrual cycle and the underlying causes for what they experience during menstruation. The existing national curriculum in Bangladesh does not address the gaps in menstruation information adolescents are receiving at home. Bangladeshi national curriculum textbooks for classes 6 to 10 (age 11–17 years) mention physical and mental changes among adolescent boys and girls during puberty, including dietary requirements and personal hygiene. However, there is only a cursory explanation of menstruation, stating that it is one of the many biological changes among girls upon reaching puberty. The national education curriculum textbooks lack practical information about how girls can adjust to the changes in their bodies during menstruation, thus missing the opportunity to create constructive discussion regarding menstruation and healthy menstrual practices [36]. In co-education schools, teachers often skip chapters containing information on menstruation, encouraging students to read it at home, thereby exacerbating the stigma already present [37]. The requests from students in our study for more comprehensive explanations of menstruation and practical advice on how to manage it demonstrate how current sources of menstruation information are inadequate.

The other key cross-cutting theme in our study was the ‘Am I normal?’ questions regarding the timing of menarche, irregular menstruation, menstrual flow heaviness, and presence of vaginal discharge. These types of questions are common among adolescents experiencing puberty across contexts [38]. However, in contexts where education about menstruation is greatly limited yet the ability to conceive is strongly linked with social inclusion, fears about possible menstrual problems may be amplified and this requires further research. In Bangladeshi society, married women have a stable position within the family and are considered active members of the marital family only after childbirth [39]. Conversely, women perceived to be infertile may be subjected to social rejection and marginalisation from their communities and families, and physical and emotional abuse at home, contributing to psychological distress [40]. Prior studies have shown how Bangladeshi women equate good health and fertility with the regularity of menstrual flow [41,42]. Menstruation education that emphasises the broad range of ‘normal’ experiences and highlights how menstrual cycle irregularity is common for the first few years after menarche is important for addressing the concerns raised by students in our study. We posit that cursory information sessions or printed materials that present average cycle lengths without sufficient elaboration on normal variations may cause undue concern when a girl’s personal experience does not match those expectations.

A large proportion of the questions submitted during our pilot study related to menstrual symptoms, primarily experiences of pain and methods of pain management. Indications within the submitted questions that girls had heard they should not take pain relief medicine for menstrual cramps is consistent with studies in other countries such as Uganda [43] and Malawi [44] that have reported concerns that taking pain medication for menstrual cramps could negatively affect a girl’s health and fertility. Menstrual pain is a significant concern of schoolgirls but has been under-addressed by menstrual health and hygiene programming [45] and under-researched in LMICs [46], where attention has been more narrowly focused on the impact of poor access to sanitation facilities and reliable menstrual materials on girls’ attendance at school [47,48]. Notably, questions about menstrual pain and its management were substantially more common in our study than questions about menstrual hygiene practices. The majority of these submissions were from rural schools. This is consistent with recent study findings from India and Bangladesh [8,49,50,51] that the majority of rural girls experienced menstrual discomforts such as cramps, backache, and particularly, lower abdomen pain. Almost 64% of girls reported missing school during menstruation due to cramps/bad physical feelings in a baseline survey before intervention in Netrokona District, Bangladesh [52]. Cross-sectional studies in India found that abdominal pain during menstruation was one of the main reasons for school absenteeism among rural adolescents, and only a small number of rural girls discussed their menstrual problems with anyone (e.g., mothers or healthcare providers) [53,54]. Our findings demonstrate an unmet need for guidance on effective pain relief for school-going girls in this context.

Most of the ‘Menstrual physiology’ submissions were from the two urban high schools, asking for the cause/need/definition for menstruation, the process of menstruation, and the link between menstruation and reproduction. Similarly, a cross-sectional study in an urban Indian community with 88 adolescent respondents reported that only 17% of girls had correct knowledge of the organ for menstrual bleeding, and about a third had knowledge that menstruation is a physiological process, while over half did not know about the cause of menstruation [55].

Students’ requests in our study for clarifications about behavioural proscriptions during menstruation highlight the need for menstruation education that acknowledges students’ social, cultural, and religious contexts. Bangladesh is a Muslim-majority country, and menstruating women are considered impure to undertake rituals within Islam [56]. Further, many of the questions requested clarifications about food restrictions which are common in Bangladesh [36,45,57].

All submissions labelled as menstrual stigma, fear and social support were from urban schools. The ‘mental suffering’ described by a student in our study was related to adults not wanting to listen to adolescents’ requests or not providing appropriate answers. In Bangladesh, around 80% of schoolgirls obtain menstruation information from mothers, grandmothers, and sisters, and approximately 36% obtain information from schools [58]. Globally, despite being the primary source of information regarding menstruation, mothers often present a negative message and share very little about their own menstrual experiences due to cultural taboos or religious proscriptions, inadequate knowledge, and embarrassment [7,59,60]. Indian studies found that parents encountered difficulties to talk to their adolescent daughters about menstrual and reproductive health [61,62]. Possible reasons for parental reluctance involved a lack of understanding and awareness about healthy menstruation, lack of communication skills, feelings of embarrassment and stigma, the generation gap between adolescents and parents, strict behaviour and tough demeanour of parents, having a conservative mindset, disinterest, and hesitancy to discuss a ‘taboo topic’ [62]. Adolescent girls also face reluctance from schoolteachers—female, and more often, male teachers have reported feeling awkward talking about this topic, displaying a negative attitude [61].

### 4.1. Strengths and Limitations

Anonymous question boxes have been used in school-based puberty and sexual education programs in high-income countries such as Australia [63], Canada [64,65], and the UK [66], in a research study assessing students’ knowledge of HIV and AIDS in Malawi [67], and in a youth-friendly sexual and reproductive health service centre in Bangladesh [68]. Such programs have shown how question boxes enable discussion of topics that planned lessons may have missed or that require further clarification and yield additional insight by capturing students’ spontaneous questions that they may have otherwise not asked due to fear of being stigmatised, punished or ridiculed [63,64]. The approach also allows instructors more time to prepare answers which may better ensure they are comfortable providing age-appropriate responses [65]. The study team also prepared answers for the students’ questions (specifically regarding menstrual symptoms and menstrual physiology, including the female reproductive system) and delivered them through the teachers. The Question Box approach complements the research methods that have conventionally been used to explore adolescents’ experiences of menstruation in Bangladesh (e.g., surveys, in-depth interviews, and focus group discussions). Conventional methods elicit responses based on prompts and require participants to discuss sensitive topics with researchers and participants are less often encouraged to ask questions themselves.

Students submitted questions anonymously, so we could not differentiate data according to sex or age but only by class levels and geography. We did not record when individual questions were submitted over the 6-month pilot period, so we were unable to explore whether students’ questions changed as their teachers covered specific topics in education sessions. Some students asked questions directly to their teachers during sessions, so the questions presented in this analysis may not represent every question students had. Questions on other aspects of puberty that were not directly menstruation-related such as those focusing on sexuality and boys’ physical development during adolescence were beyond the scope of analysis for this paper. However, these topics are inherently connected and students’ desire for education on broader sexual and reproductive health issues warrants further investigation. We have provided a clear description of the study setting and the broader context of the pilot study and this sub-study so that the readers may judge how transferable the findings may be to their own context.

### 4.2. Ethical Considerations

Generally, in a school setting, children are approached as a group, and the research participation itself may occur in a group context [69] and classrooms are also considered public places [70]. Group context can have implications for confidentiality [71]. In our study, the presence of peers could have influenced the submissions. Students broke the Question Box in some class levels in all schools, which could have led to other students not submitting questions at various points during the 6-month pilot. This may have caused a breach of confidentiality for the participants. It may have also prevented us from getting a complete presentation of the aspects of menstrual health information that adolescents wanted to know. For example, there were no questions in our dataset from classes 5–7 in one rural high school or from classes 5 and 10 from the other rural high school and one urban school.

## 5. Conclusions

Our study provides insight into the understanding of menstruation among Bangladeshi adolescent students. Analysing the content of students’ anonymous questions submitted during a pilot menstrual hygiene management intervention in Bangladeshi schools enabled us to assess the types and breadth of menstruation-related information that students desire. Bangladeshi students in our study schools wanted to obtain comprehensive information from knowledgeable teachers about the menstrual cycle including detailed explanations for the causes of menstrual variations, irregularity and symptoms, recommendations for effective hygiene and pain management practices, and advice on whether they should abide by behavioural proscriptions to avoid negative consequences to their health and fertility.

Findings from this paper will help inform future school curriculum designs and programs for providing age-specific menstruation information, with attention to the timing of menarche among Bangladeshi adolescent girls. In Bangladesh, adolescent-friendly health corners (AFHCs) were established in selected government health facilities at the district and union levels beginning in 2015. These AFHCs provide information, counselling, and services for boys and girls aged 10–19 years regarding menstruation, puberty, and sexual and reproductive health among other relevant topics. An early qualitative assessment of the program found that some adolescents learned about AFHCs and their services from their schoolteachers, highlighting the role of the school in raising awareness and linking adolescents to the health system [72]. The Bureau of Health Education under the Directorate General of Health Services set up the School Health Program at schools and madrassas where health inspectors organise and conduct monthly health education sessions for primary and secondary school students [73]. Strengthening these programs by incorporating menstruation education and promoting them through teachers would also be beneficial for adolescents.

We recommend strengthening engagement among schools, parents, and healthcare providers to address the possibility of girls receiving conflicting information and to assuage adolescents’ worry over whether their menstrual experiences are ‘normal’ or indicative of problems. Education that emphasises the variations in normal menstrual experiences could help adolescents and their caregivers better determine when a healthcare provider should be consulted. The Directorate of Primary Education and Directorate of Secondary and Higher Education should ensure that these discussions are mandatory at primary and secondary schools in Bangladesh and support the dissemination of puberty education curricula that have been successfully piloted. Further, the incorporation of menstrual health and hygiene indicators into national-level monitoring frameworks (e.g., health, education, WASH, and gender policies) is critical to assess progress.

## Figures and Tables

**Figure 1 ijerph-19-10140-f001:**
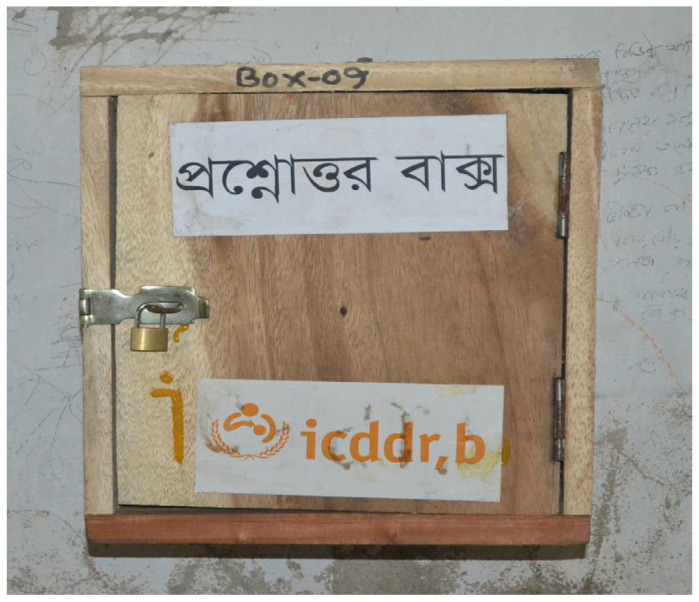
Wooden ‘Question Box’ used by adolescent school students for anonymous question submission during a 6-month pilot study in four schools in Bangladesh between August 2017 to April 2018. The wooden box cost BDT 360 (USD 4.12). It had a slit in the top and was kept under lock-and-key in all classrooms where teachers conducted puberty education sessions, as part of a menstrual hygiene management intervention at the four schools in rural and urban Bangladesh. Students were encouraged to anonymously submit general puberty and menstruation-related questions into the wooden box if they felt uncomfortable asking aloud during class.

**Table 1 ijerph-19-10140-t001:** Menstruation-related questions submitted by students in four schools in Bangladesh 2017–2018, reported by category/sub-category.

Category/*Sub-Categories*		
	Total N	Percentage (%)
Experiences of menstrual bleeding	132	35
*Timing of menstruation*	*100*	26
*Menstrual blood flow*	*14*	4
*Request for menstrual hygiene management advice*	*18*	5
Menstrual symptoms and management	122	32
Menstrual physiology	70	19
Behavioural prescriptions and proscriptions during menstruation	22	6
Concerns regarding vaginal discharge	14	4
Menstrual stigma, fear, and social support	15	4

**Table 2 ijerph-19-10140-t002:** Number of menstruation-related questions submitted by students in the urban and rural schools in Bangladesh 2017–2018, reported by category/sub-category and class level.

Category/*Sub-Categories*	Urban					
	Class Level					
	Class 5	Class 6	Class 7	Class 8	Class 9	Class 10
Experiences of menstrual bleeding	6	10	9	10	21	2
*Timing of menstruation*	*4*	*7*	*7*	*7*	*11*	*2*
*Menstrual blood flow*	*1*	*1*	*1*	*1*	*1*	*0*
*Request for menstrual hygiene management advice*	*1*	*2*	*1*	*2*	*9*	*0*
Menstrual symptoms and management	1	2	6	18	5	4
Menstrual physiology	2	5	2	15	12	4
Behavioural prescriptions and proscriptions during menstruation	2	5	2	5	3	0
Concerns regarding vaginal discharge	0	3	0	1	3	1
Menstrual stigma, fear, and social support	3	6	2	0	2	2
**Category/*sub-categories***	**Rural**					
	**Class level**					
	Class 5	Class 6	Class 7	Class 8	Class 9	Class 10
Experiences of menstrual bleeding	0	0	35	31	8	0
*Timing of menstruation*	*0*	*0*	*31*	*25*	*6*	*0*
*Menstrual blood flow*	*0*	*0*	*3*	*4*	*2*	*0*
*Request for menstrual hygiene management advice*	*0*	*0*	*1*	*2*	*0*	*0*
Menstrual symptoms and management	0	0	39	39	8	0
Menstrual physiology	0	2	12	6	6 (Class 9), 3 (Class 9–10) *, 1 (Class 10)	
Behavioural prescriptions and proscriptions during menstruation	0	0	0	1	4 (Class 9–10) *	
Concerns regarding vaginal discharge	0	0	3	2	1	0
Menstrual stigma, fear, and social support	0	0	0	0	0	0

* The questions were collected together for classes 9–10 in one rural school.

**Table 3 ijerph-19-10140-t003:** Examples of menstruation-related questions submitted anonymously by students in four schools in Bangladesh 2017–2018, classified by categories and sub-categories.

Category/*Sub-Category*	Examples of Questions Asked
Experiences of menstrual bleeding: *Timing of menstruation*	My age is 13 years but why hasn’t my period started? Will my irregular period become normalised or regular when I reach 15 years of age? Why is my period lasting for more than seven days? Period occurs twice in a year but lasts for 30 days, is it okay? Is it a health problem if girls have periods every 3–4 months, and will it be a problem for them to conceive?
Experiences of menstrual bleeding: *Menstrual blood flow*	Is there any harm in low bleeding during menstruation? What to do if there is excess blood? Why is there excessive bleeding a lot of times? Why do girls have blood clots or watery bleeding during periods?
Experiences of menstrual bleeding: *Requests for menstrual hygiene management advice*	Is it better to use a cloth or sanitary pads? If we use a cloth, how many days after should we change it? We want to know how we can maintain cleanliness while using cloth during periods. What do I do if I have menstruation in class and stain my clothes (school uniform)? How frequently should we change the cloths or pads during periods? The water supply in our washroom is not good and the washroom is dirty.
Menstrual symptoms and management	Why do girls have abdomen cramps during menstruation? How can we decrease cramps during periods? Why can’t we take medication to reduce abdominal cramps during menstruation? Why do I have acne during menstruation? Why do you have back pain during menstruation?
Menstrual physiology	What is menstruation and what is the reason for it? Why does menstruation take place? What happens when you are menstruating? Why can’t girls get pregnant without menstruation? What would happen if menstruation didn’t occur? If girls don’t menstruate are there any problems?
Behavioural prescriptions and proscriptions during menstruation	Why abstain from sex during menstruation? Why should we be cautious during menstruation? Why aren’t you allowed to eat sour foods during menstruation? Why don’t our father and mother let us go outside during menstruation? How should we conduct ourselves during menstruation?
Concerns regarding vaginal discharge	Why do girls have white discharge? What is the white substance that girls have? Why does white discharge occur? How long does it last and at what age does it start? What causes excessive white discharge? Is it genetic? What is the cure for this disease? Is there any problem if there is more white discharge? What to do if there is excess of it?
Menstrual stigma, fear, and social support	I am very scared about menstruation, what can I do to get out of this fear? Menstruation occurs in all girls but still why do girls laugh at menstruation? We talk about these issues with our older sisters, so I think we need to inform our mother about these issues. Can we tell our brother about menstruation? We need a madam (female teacher) for the subject of physical (puberty and sex) education.

## Data Availability

The authors confirm that the data supporting the findings of this study are available within the article and its Appendix A.

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
