# Peer review of "A Qualitative Content Analysis of Rural and Urban School Students’ Menstruation-Related Questions in Bangladesh"

_ijerph, 2022, doi:10.3390/ijerph191610140_

Round 1

Reviewer 1 Report

Dear Editor:

Thanks for the opportunity to review the manuscript: “We want to know”: A qualitative content analysis of rural and urban school students’ menstrual health questions in Bangladesh. This is a fascinating study and in line with the journal’s scope. I have some question which are outlined below. 

1.       The date of conduction (Data collection) should be stated in abstract.

2.       Please explain about the reasons and interests of principal investigator in the research topic.

3.       How do you ensure trustworthiness in this qualitative research? Please explain more.

4.       Finally, can the results of study be generalized? Can the results be applied to the local population, or in your context? What is your suggestion for further studies?

Reviewer 2 Report

This is an interesting paper that may help organizing puberty and sexual education in low- and middle-income countries. The paper needs some revisions, however.

Please consider the title. The questions are not only of health, but menstruation in general. And “We want to know” is too general; I suggest either to delete it or finding a more specific expression.

The abstract should mention that the answers are from girls AND boys.

The abstract says 374 submissions on menstruation, but the number of all questions is 834. It should be mentions both in the abstract. The content of the sentence on uncomfortable verbalizing should be in the beginning of abstract as it is the aim of the study. The aim of the study has not been expressed. 

Aim of the study should be mentioned also in the end of introduction.

The method section lacks important information on the number of students in these schools as well as in rural and urban schools separately. The reader needs some understanding from how large of a population these questions are drawn.

The description of qualitative analysis seems a bit too thorough. I suggest deleting both figure 2 and 3 because the process is described better in the text and the figures does not make it any clearer. These are basic stuff, not needed.

The results section is also very detailed. The first paragraph mentions rural 53% and urban 47%. Because we have no idea of the number of students in these schools, I suggest deleting the percentages here and using numbers instead, if the authors consider it relevant. Describing rural-urban differences, the percentage distributions for both areas should be presented, not numbers. This is the only way to see if some topics are more common in one of the areas. A separate table on urban rural differences is the best choice, not presenting them in the combined table 1.

 In Table 1, I suggest combining the classes because the numbers are very small in most classes and there are combined numbers already in menstrual physiology and the reader does not get a good picture on such messy table.  And the class differences are not mentioned in the text.

There are some interesting findings among the class differences, e.g., classes 7-8 have more questions on menstrual symptoms and timing which means that responding girls are at the age when menarche is expected or is fresh. Those class differences in table 1 that are important, should be mentioned in the text. I further suggest using the percentage distribution of questions when describing what questions have been reported. The numbers are more difficult to follow. Table 2 would only have one column of percentage and one column of numbers.

I suggest that these few important things are mentioned in the text as percentages.

In Discussion and particular in Implications for future practice, the above mentioned  age-specific questions needs to be taken into account. For future practice this means e.g., that programmes and education material should be age-specific reflecting the timing of menarche at that country. This is lacking from the manuscript.

The cross-cutting themes is not actual analysis but conclusions of the analysis, why it should be deleted from Results. It is discussed already but this could be emphasized in Discussion more.

The discussion is organized strangely. A good habit is to give first the main results of the study.

The discussion of the Question box validity cannot start the discussion. This should be together with other method discussion, wherever that will be placed.   

There is a lot of text in discussion. I suggest the authors to consider what is relevant concerning their own results and what is more general what can be deleted.

Round 2

Reviewer 2 Report

The text has improved but I still found two things that needs be addressed.

-          The question in the title “Is it a problem” is not appropriate. What is “it”? I suggest delete this question or if the authors still insist to keep it, then specify what “it” is.

-          The discussion needs some reorganization. The new long text on Boxes does not belong to Implications for future research but to the subsection of Limitations. There is actually no implications for future research in the text, why  the title Implications for future research and practice is not needed and because those practice things are better situated in Conclusions and recommendations.

Author Response

We would like to thank the reviewer for providing further comments and suggestions. We have replied (in italics) to the reviewer’s comments (cited in normal font) and revised the manuscript.

Point 1: The question in the title “Is it a problem” is not appropriate. What is “it”? I suggest delete this question or if the authors still insist to keep it, then specify what “it” is.

Response 1: Thank you for this suggestion. We have deleted the quote and now the title is "A qualitative content analysis of rural and urban school students’ menstruation-related questions in Bangladesh."

Point 2: The discussion needs some reorganization. The new long text on Boxes does not belong to Implications for future research but to the subsection of Limitations. There is actually no implications for future research in the text, why  the title Implications for future research and practice is not needed and because those practice things are better situated in Conclusions and recommendations.

Response 2: We have reorganized the Discussion. The information on the Question Box approach Anonymous question boxes…” is now under Strengths and Limitations. Additionally, the Findings from this paper...” paragraph is now under Conclusion and recommendations. As suggested, we have deleted the paragraph on Implications for future research and practice.